# Wheelchair Rugby Sprint Force-Velocity Modeling Using Inertial Measurement Units and Sport Specific Parameters: A Proof of Concept

**DOI:** 10.3390/s23177489

**Published:** 2023-08-29

**Authors:** Marc Klimstra, Daniel Geneau, Melissa Lacroix, Matt Jensen, Joel Greenshields, Patrick Cormier, Ryan Brodie, Drew Commandeur, Ming-Chang Tsai

**Affiliations:** 1School of Exercise Science, Physical and Health Education, University of Victoria, Victoria, BC V8P 5C2, Canada; 2Canadian Sport Institute Pacific, Victoria, BC V9E 2C5, Canada; 3Canadian Sport Institute Ontario, Toronto, ON M1C 0C7, Canada; 4Wheelchair Rugby Canada, Ottawa, ON K1G 4K3, Canada

**Keywords:** wheelchair, IMU, sport

## Abstract

Background: Para-sports such as wheelchair rugby have seen increased use of inertial measurement units (IMU) to measure wheelchair mobility. The accessibility and accuracy of IMUs have enabled the quantification of many wheelchair metrics and the ability to further advance analyses such as force-velocity (FV) profiling. However, the FV modeling approach has not been refined to include wheelchair specific parameters. Purpose: The purpose of this study was to compare wheelchair rugby sprint FV profiles, developed from a wheel-mounted IMU, using current mono-exponential modeling techniques against a dynamic resistive force model with wheelchair specific resistance coefficients. Methods: Eighteen athletes from a national wheelchair rugby program performed 2 × 45 m all-out sprints on an indoor hardwood court surface. Results: Velocity modelling displayed high agreeability, with an average RMSE of 0.235 ± 0.07 m/s^−1^ and *r*^2^ of 0.946 ± 0.02. Further, the wheelchair specific resistive force model resulted in greater force and power outcomes, better aligning with previously collected measures. Conclusions: The present study highlights the proof of concept that a wheel-mounted IMU combined with wheelchair-specific FV modelling provided estimates of force and power that better account for the resistive forces encountered by wheelchair rugby athletes.

## 1. Introduction

Force-velocity (FV) profiling is a commonplace assessment in able-bodied sports [1,2,3]. This performance modeling technique is a valuable way to predict the strength and speed capabilities of athletes from specialized tests [3,4]. This includes both a vertical profile based on variations of loads during lower body movements, such as the jump squat, as well as a horizontal force velocity profile based on maximal overground sprint performance [5,6,7]. The outcomes of these protocols have been effective at determining athlete performance strengths and deficiencies, providing useful information to prescribe training programs [8,9]. Recently, FV investigations have been applied to wheelchair sport athletes [10,11,12]. For example, a laboratory-based wheelchair force-velocity protocol has been developed based on sprints of varied resistances on a wheelchair ergometer [10,13]. This, for the first time, supports a novel application of the FV modeling approach in wheelchair athletes and can help set the standard for the estimation of the optimal force and velocity capability of these athletes and potentially be used as a common assessment for training, competition readiness, and athlete classification [14,15]. However, a limitation of this laboratory-based approach is access to an instrumented ergometer and, therefore, this testing may not be preferred to court-based performance tests. As the ability to accelerate and maintain high speed are critical elements of game success in wheelchair sport, a common assessment is sprinting, typically done over a 20 m distance [16]. Recently, the application of inertial measurement units (IMUs) affixed to sport wheelchair frames and wheels has provided the opportunity to measure accurate wheelchair kinematics during training, testing, and competition [12,14,15,17,18,19]. In fact, using the angular rate from the gyroscope in wheel-mounted sensors has been shown to provide accurate wheelchair speed measurements [17,19]. The common use of sprinting as an assessment in wheelchair sports, paired with the novel use and accurate wheelchair speed measurement from IMUs, presents a unique opportunity to apply sprint-based force and velocity modeling in wheelchair sport, similar to the horizontal FV approach used in able-bodied athletes [5,11].

The current FV sprint model, as applied to able-bodied athletes during sprinting, uses a mono-exponential function with kinetic components of inertial force and air resistance [4,5]. This model has been verified from historical and current literature investigating sprinting athletes and represents both experimental and theoretical explanations of change in kinematics and kinetics during sprinting [20,21]. This exponential increase in speed, as modeled by the mono-exponential function, is common in many acceleratory movements, including wheelchair sprints, and could be a useful function with which to define the parameters of theoretical maximal speed and force for wheelchair athletes similar to able-bodied athletes [11]. Recently, a sprinting based force velocity profile has been performed on short sprints on a court (20 m) as well as on an ergometer with wheelchair basketball athletes [11]. Brassart et al. (2023) included wheelchair-resistive force in their model from a deceleration (roll down) task [11]. The incorporation of a roll-down task has also been included in other investigations in wheelchair tennis to estimate resistive forces [22]. This test is important as it takes into account critical resistive variables encountered in wheelchair sport such as the effect of system mass (athlete and wheelchair), internal bearing, and rolling resistance of all wheel and roller hubs as well as differences in air resistance coefficients [13,22,23,24]. Further, each wheelchair style, between and within each wheelchair sport, will have specific differences that need to be accounted for [25]. While the current mono-exponential model has great potential to fit wheelchair sprint performance, in its current form it does not include wheelchair-specific parameters and it is unknown whether the inclusion of these variables would impact the estimation of these FV metrics over the current model. While the resistance during wheelchair sport can be quantified using roll-down tests, there is uncertainty about their accuracy as roll-down methodologies often imply a constant resistance factor to wheelchair propulsion, while other work indicates that a more dynamic resistive force may be present [22,24,26]. Additionally, roll-down tests are not always possible or performed consistently during regular performance assessment, which may result in misrepresentation of the true wheelchair resistance [22]. Interestingly, valuable work has been carried out to develop equations to estimate the parameters of force required during wheelchair movement without the necessity to perform individual roll-down tests [13] . Particularly, Chua et al. (2010) used a wind tunnel as well as strictly controlled roll-down tests on different surfaces to create a dynamic velocity-dependent function to estimate the inertial, drag, gravitational, and lift forces of a wheelchair rugby chair across varied velocities and surfaces [13]. Based on this work and the resulting equation, there is great potential to use this equation to augment the current monoexponential sprint horizontal FV profile with wheelchair performance-specific tuned parameters and to potentially remove the requirement of a roll-down test to estimate resistive force. However, while this type of modification is an important first step in improving force estimates for wheelchair sport sprint FV profiling, there are substantial sport-specific and individualized equipment modifications that can change the force required to push a wheelchair, and further work will be required to optimize this modeling approach [10]. 

Therefore, the purpose of this study is to compare wheelchair rugby sprint FV profiles developed from a single wheel-mounted IMU using the current mono-exponential model as developed by Samozino et al. (2016) against an augmented model with wheelchair rolling resistance coefficients developed by Chua et al. (2010) [4,13]. We hypothesize that the wheelchair sprint FV model will result in significantly higher force and power estimates than the current model, thereby providing a more comprehensive estimate of wheelchair athlete kinetics. This work will support a proof of concept of the use of IMU wheelchair speed estimates, alongside wheelchair rugby-specific FV modeling, to develop wheelchair rugby-specific FV profiles. 

## 2. Materials and Methods

### 2.1. Participants and Testing Protocol 

Eighteen participants (M = 17, F = 1, mass = 69.51 ± 12.08 kg, chair mass = 18.15 ± 1.15 kg, system mass = 87.66 ± 12.59 kg, chair height = 119.05 ± 5.76 m) from a national wheelchair rugby program were included in this analysis. Following a standard warm-up, subjects performed two indoor 45 m sprint trials on a standard hardwood playing surface. Forty-five meters was chosen as the shortest distance when athletes achieved a plateau in velocity as required for the mono-exponential model fit. For each trial, athletes were instructed to attempt to accelerate and reach their maximum velocity as quickly as possible within the 45 m distance. A 2–3-min rest period was provided between trials to ensure subject maximum effort could be achieved. One Xsens™ dot IMU (60 Hz) was mounted on the athlete’s right wheelchair wheel hub using a custom-made mount such that the positive *z* and *x* axes projected radially within the plane of wheel rotation and the y axis was orthogonal to the wheel rotation (Figure 1a). 

### 2.2. Data Analysis

#### 2.2.1. IMU Velocity Estimation and Sprint Detection

Gyroscope Y data were collected from all athletes during sprint trials using data logging on sensors using the Xsens™ dot mobile application. All data were analyzed using custom written Python™ (Python 3.7, Beaverton, OR, USA) software. Sprint onset using the gyroscope data was determined by the first instance of the signal greater than 10 °/s, indicating a significant rotation of the wheel. Once detected, gyroscope data were converted from °/s to radians/s to calculate translational velocity using the fundamental angular kinematic formulae outlined in Equations (1) and (2). Velocity data were then passed through a 5th order 4.5 Hz lowpass Butterworth filter. Sprint trials were cropped to the subject maximum translational velocity, in accordance with previous horizontal FV research [3].
(1)$$\omega =\frac{\pi }{180}\;\cdot \frac{\theta }{s}$$
(2)$$V_{t}=\omega \;\;\cdot \;W_{radius},$$
where *V_t_* is the translational velocity, $W_{radius}$ is the wheel radius, θ is the wheel angle in degrees, and ω is the angular velocity in radians per second.

#### 2.2.2. Velocity Modelling

All sprint trials were time-normalized over 1000 samples. Each trial was then fitted to a velocity model and subsequent acceleration model as outlined in Equations (3) and (4).

Equation (3). Velocity
(3)$$V_{t}=V_{max}\left(1-e^{-t+t_{0}/\tau }\right)$$

Equation (4). Acceleration
(4)$$a_{t}=\frac{V_{max}}{\tau }\left(e^{-t+t_{0}/\tau }\right),$$
where $a_{t}\;$ is the translational acceleration, $V_{max}$ is the maximum velocity reached during the sprint trial, *t* is time, *t*_0_ is time offset, and τ is the acceleration time constant. 

#### 2.2.3. Resistance Force Modelling

Using the velocity and acceleration derived from Equations (3) and (4), either a commonly used air friction resistance force model (Section Air Friction Resistive Force Modelling) or a chair rolling friction resistance force model (Section Chair Rolling Friction Resistive Force Modelling) was used to calculate horizontal FV output measures using the methods outlined by Morin et al. (2019). Specifically, theoretical maximum force ($F_{0}),$ theoretical maximum velocity ($V_{0})$, decrease ratio of force (DRF), maximum power ($P_{Max}$), and force velocity slope ($F_{Slope}$) were calculated using both models outlined below. 

##### Air Friction Resistive Force Modelling

The first model used to determine resistive force was termed the air friction (AF) resistive force modelling technique, as this approach uses air friction based on athlete stature during performance as the prevalent resistant forces during sprinting. The AF approach was performed using FV methods outlined in previous works for overground sprinting using Equations (5)–(8) [4,5,27]. 

Equation (5). Athlete surface area
(5)$$SA=0.2025\;\cdot \;H^{0.725}\cdot \;m^{0.425}\cdot 0.266$$

Equation (6). Air density
(6)$$\rho =1.293\cdot \frac{Pressure}{760}\cdot \frac{273}{273+T}\;$$

Equation (7). Drag coefficient
(7)$$k=\frac{1}{2}\rho \cdot SA\cdot 0.9\;$$

Equation (8). Air resistive force (AF)
(8)$$F_{R\;}=-k\;{\left(V_{t}-V_{wind}\right)}^{2}\;,$$
where *H* is the athlete chair stature during performance, m is system mass, pressure is the barometric pressure measured in hectopascal, *T* is temperature measured in Celsius, and $V_{wind}$ is the wind velocity. As all sprints were performed indoors, wind velocity is assumed to be zero.

##### Chair Rolling Friction Resistive Force Modelling

The second technique used to model resistive force was the chair rolling friction resistive force (CRF), which accounted for wheelchair-specific force included drag, lift, and rolling friction components [13]. The chair rolling friction model (CRF) was calculated based on the work done by Chua et al. (2010) [13]. This included a dynamic friction constant μ dependent on wheelchair instantaneous acceleration and velocity measures as outlined in Equations (9)–(12). 

Equation (9). Air drag constant
(9)$$C_{D}=\frac{0.2955}{V_{t}+0.0762}+0.1608$$

Equation (10). Air lift constant
(10)$$C_{L}=\frac{0.2099}{V_{t}+0.1218}+0.1666$$

Equation (11). Dynamic friction coefficient
(11)$$\mu =-\frac{F_{I}+F_{D}}{F_{G}+F_{L}}\;\;,\;\;\;-\frac{-am+C_{D}V_{t}{}^{2}}{gm+C_{L}V_{t}{}^{2}}$$

Equation (12). Chair resistive force (CRF)
(12)$$F_{R}=\mu mg\;,$$
where $F_{I}$ is the inertial force, $F_{D}$ is the drag force, $F_{G}$ is the force of gravity, and $F_{L}$ is the lift force. The acceleration due to gravity, g, was assumed to be $9.81\;m/s^{2}$. 

For each resistive force model, the calculated force was added to the net force equation as outlined in Equation (13) below: (13)$$F_{Net}=ma_{t}+F_{R},$$
where $F_{Net}$ is the net force in the horizontal, translational direction, and m is system mass, and $a_{t}$ is the modelled translational acceleration.

### 2.3. Statistical Analysis

First, to determine the agreeability of the mono-exponential model fit velocity data obtained from a single wheel-mounted IMU in wheelchair sprints, goodness of fit statistics (RMSE, $r^{2}$) were used. Secondly, to compare the differences between the AF and CRF resistive force models, paired t-tests were used to assess statistical differences between FV measure outputs of each model ($P_{max}$, $F_{0}$, $V_{0}$, DRF, FV Slope). Further, FV measure outputs of each model were also qualitatively compared to previously collected wheelchair FV values from other recent studies [10]. To support a qualitative comparison, previously collected values were adjusted based on mean maximum velocity values in each group. All dependent FV measures were adjusted based on this correction. 

## 3. Results

### 3.1. Velocity Modelling Goodness of Fit

Figure 2 shows the average velocity and velocity model fit for all participants and all sprint trials modeled using Equation (3). Velocity modeling reported high agreeability with the measured velocity data with an average RMSE of 0.235 ± 0.07 m·s^−1^ and $r^{2}$ of 0.946 ± 0.02. These results indicate that the velocity model adequately represents the measured velocity data for analysis.

### 3.2. Resistance Force Model FV Measures Comparison

Table 1 includes a summary of horizontal force velocity measure outputs from AF and CRF resistance models with both descriptive data (mean and standard error) and inferential comparisons. Table 1 also includes comparable FV metrics from recent wheelchair sprint investigation results. Significant differences in FV outputs between AF and CRF modeling techniques were observed. Notably, there were significant differences in theoretical maximum force ($F_{0}$), maximum power output ($P_{max}$), decrease ratio of force (DRF), and force-velocity slope (FV Slope). No significant differences in calculated theoretical maximum velocity ($V_{0}$) were observed. Qualitative comparisons show comparable $P_{max}$, $F_{0}$ from previous research to CRF from the present investigation. 

Figure 3 shows the average FV profiles developed using both the CRF (red) and AF models (blue) as well as each athlete’s individual FV profile for CRF (orange) and AF (light blue). Figure 4 shows the average, individual athlete data and interquartile range for metrics ($P_{max}$, $F_{0}$, $V_{0}$, FV Slope) developed for each of the CRF and AF models. The CRF technique consistently generates significantly higher $F_{0}$ and $P_{max}$ values when compared to the air friction resistance model, as can be seen in Figure 3 and Figure 4. This systematic difference between techniques causes cascading effects on variables, which are dependent on these values such as DRF and FV Slope. Further, for qualitative comparison, in Figure 4 the mean FV variables from Janssen et al. (2023) are displayed as dotted vertical lines with uncorrected values (gray) and values corrected to a common velocity (red) [10]. 

## 4. Discussion

This study investigated the potential to develop a wheelchair rugby FV model using a wheel-mounted IMU speed measurement and wheelchair rugby-specific resistance parameters. The derived performance model showed high agreement with the measured speed data and the wheelchair-specific model resulted in higher estimates of force and power over the common model, which includes only basic inertial parameters and air resistance. Overall, this demonstrates that a wheel-mounted IMU combined with a wheelchair rugby-specific FV model may provide estimates of force and power that better account for the resistive forces encountered by wheelchair rugby athletes. 

IMUs have emerged as valuable tools for enhancing performance analysis and training methodologies in wheelchair sports [14,15,17,18,22]. Specifically, important developments and validations have demonstrated that IMUs mounted on the frame and wheel can provide accurate and reliable linear and rotational kinematic measurements during wheelchair training and competition [17,19]. For example, Mason et al. (2014) demonstrated that a wheel-mounted IMU had a very small random speed error of 0.004 m⋅s^–1^ to 0.015 m⋅s^–1^ as compared to high speed video, and a reliability of less than 2% CV for peak speed measurement and less than 1% CV across all speeds [19]. These measurements were confirmed by van der Slikke et al. (2015) when comparing a three-IMU sensor configuration against optical motion capture [17,28,29]. This high level of accuracy and reliability, especially with respect to speed measurement, strongly supports the use of IMUs to provide important wheelchair mobility metrics. Indeed, IMU measurement is now commonplace in wheelchair sport analysis in basketball, tennis, and rugby [14,18]. Given the success and accuracy of wheelchair IMUs, this sensor has been suggested to support important concepts, such as classification, and is now ready to provide the measurement to investigate advanced performance modeling approaches such as FV profiling [11,15]. 

FV profiling from on-field/court sprinting, while now common in able-bodied sport, has only recently been applied to wheelchair athletes [11,22,30]. Brassart et al. (2023) performed a study comparing FV variables produced during different tasks such as a 20 m court sprint, an ergometer maximum sprint, and a horizontal ballistic push off task [11]. The analysis consisted of modeling the acceleration (first derivative of IMU speed data) from the on-court and ergometer testing to estimate the force required by the athlete. The model included rolling resistance as determined using a roll-down test from the on-court task [26]. The authors also used a polynomial fit of the IMU speed to determine wheelchair acceleration, which was then modeled to estimate force. The present investigation differed in approach, in that IMU speed was modeled using a common mono-exponential function and did not use a roll-down test but included a dynamic velocity-dependent function of wheelchair rugby-specific chair resistance force [5]. Both of these approaches are valuable and have important advantages and limitations. The use of a roll-down test to determine resistance has important benefits to individualize the resistance required under different conditions and chair installations [22]. Sauret et al. (2012) originally presented the roll-down test as a means to determine a rolling resistance coefficient of multiple different wheelchair configurations in field conditions [26]. Rietveld et al. (2021) has also demonstrated the benefit of using roll-down tests to determine a rolling resistance value and has shown significant differences in resistance due to tire pressure and surfaces in wheelchair tennis [22]. In the development of the dynamic equation used in the present study, Chua et al. (2010) also used roll-down tests [13]. An important distinction from other research, the equations from Chua et al. (2010) are dependent on velocity and are therefore dynamic, while other roll-down investigations have only resulted in linear, fixed resistive forces [22,26]. Further, while Reitveld et al. (2021) was able to use roll-down tests to effectively predict power from the roll-down tests, using roll-down regions during a 10 m sprint was not successful at estimating power during sprint tests [22]. While the roll-down test may potentially be a useful task with which to assess individualized resistance, more research is needed to verify its accuracy across different speeds [22,26]. Additionally, while there is the potential to use roll-down tests in sport assessments, these tests may not be able to be performed during large testing sessions, and require a degree of expertise to complete and analyze [22,26]. This supports the potential to use a dynamic chair resistive force equation that can account for more variables such as tire pressure and surface, when roll-down tests cannot be effectively performed or analyzed. As a major goal of this research is to provide better estimates of the force and power of wheelchair athletes using IMU speed measurements, there is a need to continue to find appropriate and accessible measurement and predictions of chair resistive force. 

As an important comparison for the magnitude of predicted force and power in the present investigation, Janssen et al. (2023) recently reported force velocity estimates of experienced wheelchair rugby athletes during an ergometer sprint against varied resistances [10]. They were able to directly measure ergometer torque and report tangential force and power. In this study, athletes produced a mean optimal power of 532 W and $F_{max}$ calculated from the athlete-specific regression equations was 343 N. These values are presented in Figure 4 as dotted gray lines as a reference against values from the present experiment. As can be observed, while the FV slope from Janssen et al. (2023) is comparable to the FV slope reported for the CRF model, the values of force and power from Janssen et al. are higher than both the AF and CRF models [10]. As the CRF model is more closely aligned to the values of Janssen et al. (2023), this may qualitatively support the results from the CRF model [10]. Interestingly, the maximum velocities achieved by the athletes on the ergometer in Janssen et al. (2023) are higher than reported in this study [10]. Given that the athletes are at a comparable level and have a similar FV slope, it may be that the ergometer resistance in Janssen et al. (2023) is not consistent with the resistance of the chairs on the wood surface in the present experiment [10]. This lower relative resistance on the ergometer may allow the athletes in Janssen et al. (2023) to achieve a higher maximum velocity [10]. Assuming that the athletes between experiments are of a similar performance level, if the velocities are normalized and the regression equations in the Janssen et al. (2023) paper are used to predict new normalized force and power, then the force and power estimates become much more closely aligned to the values predicted in this athlete cohort as seen by the red dotted line in Figure 4 [10]. Using either corrected or uncorrected values when comparing Janssen et al. (2023) and the present experiment provides potential support for the current monoexponential sprint velocity modeling with CRF inclusion [10]. Further, while it is difficult to compare between wheelchair sports due to differences in athlete ability and equipment constraints, the velocities achieved and predicted forces and powers from the athlete cohort in Brassart et al. (2023) are similar to the ones reported in this present investigation [11]. It is important to note that the basketball athletes in Brassart et al. (2023) achieved peak velocity over a much shorter distance than the wheelchair rugby athletes in the present study, where 45 m was required [11]. Overall, the force and power estimates from the CRF model in the present experiment show qualitative agreement with other studies, which shows the great potential to use this model to account for the resistive forces present during wheelchair rugby. However, while Chua et al. (2010) accounted for multiple important wheelchair rugby-specific considerations, there are many parameters that may require further consideration and inclusion in predictive modeling equations [13]. 

There are a number of technical parameters that are unique to wheelchair athletes, which alter motion and movement performance. Athlete stature and system mass have been found to alter wheelchair mobility [31,32], which is similar to traditional overground able-bodied performers. However, factors such as rolling friction, dynamic air drag and lift forces, and wheel camber angle are known physical quantities, which have often been unaccounted for in previous investigations [13,24]. While at low speeds factors such as chair lift may be negligible, due to the dynamic nature of this force, it may be a justified inclusion in the overall resistive force model [13,23]. A unique benefit of the dynamic equations presented by Chua et al. (2010) is the investigation and inclusion of many variables specific to wheelchair rugby across speeds and conditions [13]. This dynamic equation is therefore unique to wheelchair rugby and necessitates the value of reproducing and validating similar equations for other wheelchair sports. 

## 5. Conclusions

This study highlights the proof-of-concept that a single wheel-mounted IMU in conjunction with a wheelchair-specific resistive force model better estimates the force and power generated throughout a wheelchair linear sprint. However, while this approach is promising, there is a need to compare these measurements to a kinetic criterion such as a calibrated ergometer or force-instrumented wheel. Additionally, there is value in further confirmation of kinetic estimates to individualized, controlled, deceleration tests, to determine all technical aspects of wheelchair mobility that could result in differential resistance. Through more development, this modeling approach using IMUs could help support the estimation of kinetic parameters in wheelchair sport studies investigating changes in equipment and training and during competition. 

## Figures and Tables

**Figure 1 sensors-23-07489-f001:**
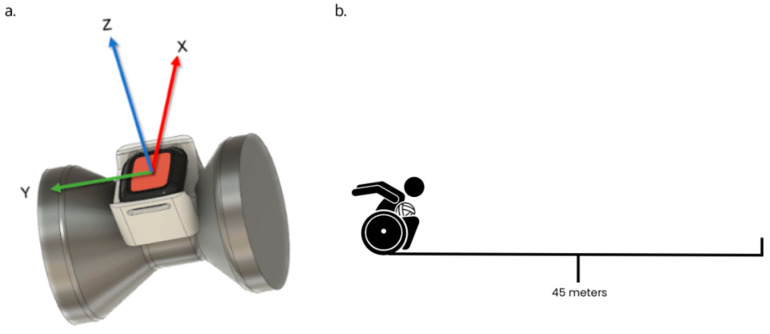
(**a**) The custom-made mount placed on the wheel hub showing the IMU axis orientation; (**b**) The schematic of the 45 m sprint trial athletes completed.

**Figure 2 sensors-23-07489-f002:**
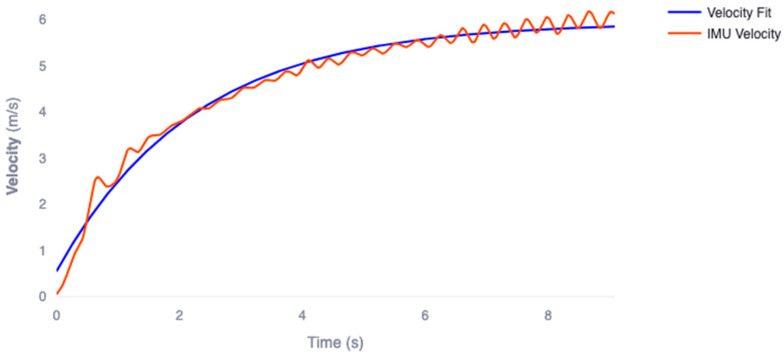
Sample trace of velocity obtained from wheel-mounted IMU (orange) compared to the calculated mono-exponential velocity model (blue) for a single participant.

**Figure 3 sensors-23-07489-f003:**
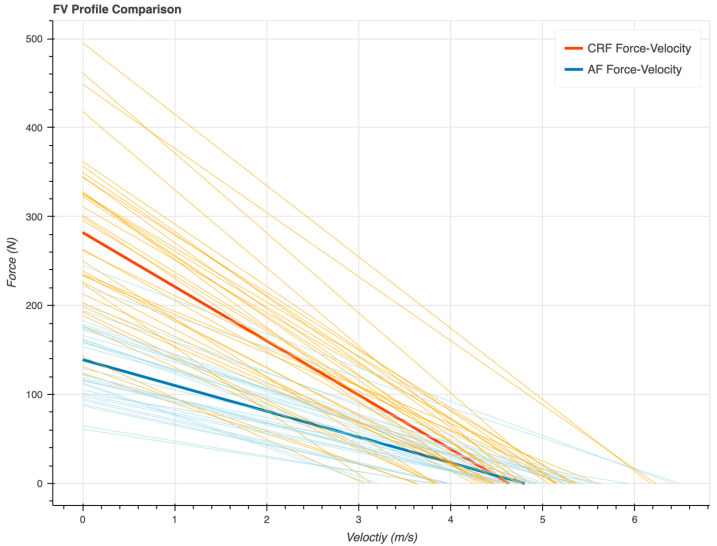
Comparison of force-velocity linear profiles between resistive force modeling techniques. Averages are displayed in the solid orange (CRF) and blue (AF) lines.

**Figure 4 sensors-23-07489-f004:**
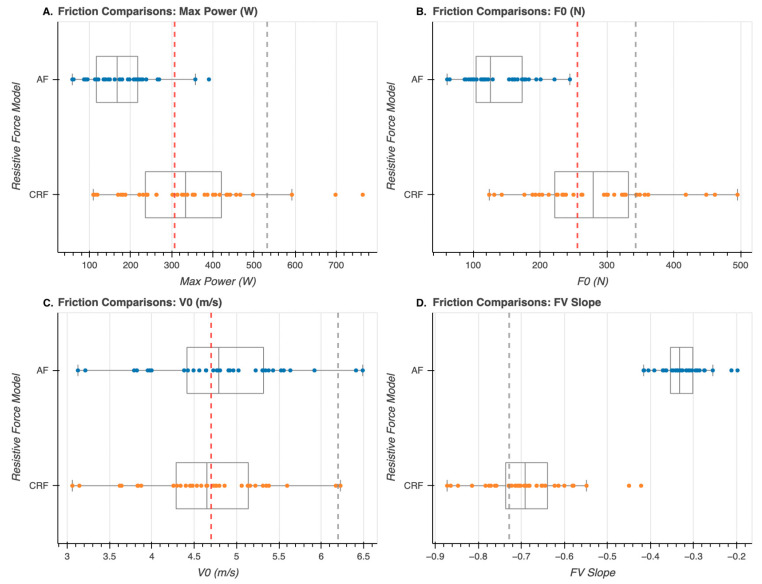
(**A**–**D**) Comparisons of sprint-derived FV variables between resistive modeling techniques. Directly measured force and velocity data averages obtained from a study by Janssen et al. (2023) [10] are represented as vertical lines in each respective graphic. Using the FV Slope information, *V*_0_ and *F*_0_ values were adjusted from their original values (gray vertical lines) to a corrected value (red vertical lines) to accommodate for velocity magnitudes between tested populations.

**Table 1 sensors-23-07489-t001:** Horizontal force-velocity measurement comparisons between resistive force and velocity modeling techniques. Significant differences in variables determined from each resistive force modeling technique are identified with an Asterix (*). Force-velocity values from other recent wheelchair sprint investigations (Janssen et al. (2023) [10], Brassart et al. (2023) [11]) are also displayed.

Model	Athlete Population		Pmax (W)	F0(N)	DRF	V0 (m/s)	FV Slope
CRF	Wheelchair Rugby: National Team	Mean	338.738 *	282.125 *	−6.794 *	4.631	−0.686 *
		SEM	25.23	15.14	0.1514	0.12	0.02
AF		Mean	172.078 *	139.102 *	−3.309 *	4.804 *	−0.327 *
		SEM	12.71	7.31	0.08	0.13	0.01
Janssen et al.	Wheelchair Rugby: Experienced	Mean	307.174	256.056	N/A	4.700	−0.728
		SEM	42.15	24.75	N/A	0.33	0.16
Brassart et al.	Wheelchair Basketball: National Team	Mean	277.6	244	N/A	4.700	N/A
		SEM	25.15	16.37	N/A	0.15	N/A

## Data Availability

Data used in this study were shared by Wheelchair Rugby Canada and are not available for further sharing.

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
