# Peer review of "Wheelchair Rugby Sprint Force-Velocity Modeling Using Inertial Measurement Units and Sport Specific Parameters: A Proof of Concept"

_sensors, 2023, doi:10.3390/s23177489_

Round 1

Reviewer 1 Report

As this study presents a proof of concept about the combination of wheel mounted IMU with a wheel chair specific FV model for the provision of estimates of force and power that better account for the resistive forces encountered by wheelchair rugby athletes. 

1. I would suggest to add a comparison table of different type of approaches adopted in the literature about estimates of force and power that better account for the resistive forces encountered by wheelchair rugby athletes.

Please compare in terms of performance with your achievements in this study. 

2. Most of the references are quite old. I would suggest to improve them by adding some latest relavent study in the literature. 

3. Most of the figures quality is not good. They are blurry which should be improved. 

English language is fine. 

Author Response

As this study presents a proof of concept about the combination of wheel mounted IMU with a wheel chair specific FV model for the provision of estimates of force and power that better account for the resistive forces encountered by wheelchair rugby athletes. 

1. I would suggest to add a comparison table of different type of approaches adopted in the literature about estimates of force and power that better account for the resistive forces encountered by wheelchair rugby athletes.

Please compare in terms of performance with your achievements in this study. 

Response: We thank the reviewer for this comment and agree that a comparison table would be a nice addition to the document. We have added a comparison table to show better comparisons in terms of performance with our study data.

2. Most of the references are quite old. I would suggest to improve them by adding some latest relavent study in the literature. 

Response: We thank the reviewer for this comment and have updated the document with some more recent studies.

3. Most of the figures quality is not good. They are blurry which should be improved. 

Response: We thank the reviewer for noticing this issue and have updated the figures.

Reviewer 2 Report

Para-sports such as wheelchair rugby have seen increased use of IMU's to measure wheelchair mobility. The accessibility and accuracy of IMU’s has enabled the quantification of many wheelchair metrics and the ability to further advance analysis such as force-velocity (FV) profiling.

However, the force velocity modeling approach has not been refined to include wheelchair specific parameters.

 The purpose of the authors was to compare wheelchair rugby sprint FV profiles, developed from a wheel-mounted IMU, using current mono-exponential modeling against a dynamic resistive force model with wheelchair specific resistance coefficients.

Eighteen athletes from a national wheelchair rugby program performed 2 x 45m all-out sprints on an indoor hardwood court surface. Goodness of fit statistics displayed high agreeability of velocity modeling, with an average RMSE of 0.235 ± 0.07 m/s^ and r^2 of 0.946 ± 0.02.

Further, the wheelchair specific resistive force model resulted in greater force and power outcomes, which more consistently aligns with previously collected measures.

Authors concluded that their study highlights the proof of concept that a single wheel-mounted IMU combined with a wheelchair specific FV model provided estimates of force and power that better account for the resistive forces encountered by wheelchair rugby athletes.

This is an interesting study.

I have some minor suggestions with a pure academic spirit.

1.       Abstract must better summarize the sections

2.       Define the acronym of IMU (inertial measurement unit)

3.       M&M avoid short paragraphs

4.       Describe the figures in details in the body of the manuscript

5.       Report formulas/equations with more care (refere to MDPI standard)

6.       Check the resolution of figures 2-4

7.       Add the conclusions

Author Response

Para-sports such as wheelchair rugby have seen increased use of IMU's to measure wheelchair mobility. The accessibility and accuracy of IMU’s has enabled the quantification of many wheelchair metrics and the ability to further advance analysis such as force-velocity (FV) profiling. 

However, the force velocity modeling approach has not been refined to include wheelchair specific parameters.

 The purpose of the authors was to compare wheelchair rugby sprint FV profiles, developed from a wheel-mounted IMU, using current mono-exponential modeling against a dynamic resistive force model with wheelchair specific resistance coefficients.

Eighteen athletes from a national wheelchair rugby program performed 2 x 45m all-out sprints on an indoor hardwood court surface. Goodness of fit statistics displayed high agreeability of velocity modeling, with an average RMSE of 0.235 ± 0.07 m/s^ and r^2 of 0.946 ± 0.02. 

Further, the wheelchair specific resistive force model resulted in greater force and power outcomes, which more consistently aligns with previously collected measures. 

Authors concluded that their study highlights the proof of concept that a single wheel-mounted IMU combined with a wheelchair specific FV model provided estimates of force and power that better account for the resistive forces encountered by wheelchair rugby athletes.

This is an interesting study.

I have some minor suggestions with a pure academic spirit.

  1. Abstract must better summarize the sections

Response: We thank the reviewer this comment and have updated the abstract to better summarize the sections, additionally we have added subheadings.   

  1. Define the acronym of IMU (inertial measurement unit)

Response: We have added a definition of the IMU acronym in the abstract

  1. M&M avoid short paragraphs

Response: We thank the reviewer for this comment, we have removed subsections and combined text to avoid short paragraphs with a general update to the M&M.

  1. Describe the figures in details in the body of the manuscript

Response: We thank the reviewer for this comment and have added better figure descriptions in the manuscript body.  

  1. Report formulas/equations with more care (refere to MDPI standard)

Response: We thank the reviewer for this comment and have checked to ensure that formulas/equations conform to the MDPI standards.

  1. Check the resolution of figures 2-4

Response: We thank the reviewer for noticing this issue. We have modified the figure.

  1. Add the conclusions

Response: We thank the reviewer for this comment and have added a conclusion paragraph

Reviewer 3 Report

Review sensors-2537379-peer-review-v1

The paper “Wheelchair Rugby sprint force-velocity modeling using inertial measurement units and sport specific parameters: A proof of concept” is interesting. One must be very careful when writing such papers. The methods and the results sections have to be compatible. Unfortunately, the authors did not take proper care. Reading the results section, I had the impression that there were no descriptions of what was calculated in the method. The exact comments are below, with the minor and major divisions.

Minor comments: there should not be a period after the title (line 4). In the Introduction section, where the names of the authors are given, there should also be reference numbers for the bibliography items (lines 67, 86, 99, 100). It needs to be corrected throughout the manuscript.

Participants section: I propose to add years of training experience and a category in the disability classification for Rugby.

Line 119 – What is the symbol next to Xsens?

Lines 139 – 140 – there should also be an explanation of θ.

Section 2.4.3 Resistance Force Modelling should be divided into two chapters, AF and CRF, because otherwise the formulas blend together.

Major comments:

Results section should be refined. I don't understand subsection 3.1 What does Figure 2 refer to? There is no reference in the text. Is this for one person? Please write using what equation the results were obtained for velocity modeling. I understand that it was equation 2a?

Using what test were the results described in Table 1 compared? What is the FV slope, and DRF? How were they counted? How was theoretical maximum force (?0), and maximum power output counted? This are information that should be described in the methods. Figure 4 is important and should be properly discussed in the text. Please also number the subplots as A, B, C and D. In the description of the figure there is a reference to the work of Janssen et al. This was not mentioned in the methods.

In conclusion, the authors need to describe the chapters material and methods and results more precisely so that they are compatible. The reader must have clarity on what results from what and what tests are being compared.

Author Response

The paper “Wheelchair Rugby sprint force-velocity modeling using inertial measurement units and sport specific parameters: A proof of concept” is interesting. One must be very careful when writing such papers. The methods and the results sections have to be compatible. Unfortunately, the authors did not take proper care. Reading the results section, I had the impression that there were no descriptions of what was calculated in the method. The exact comments are below, with the minor and major divisions.

Response: We thank the reviewer for the time and consideration of this manuscript and have updated the document based on these important points. We have answered the detailed comments below.

Minor comments: there should not be a period after the title (line 4). In the Introduction section, where the names of the authors are given, there should also be reference numbers for the bibliography items (lines 67, 86, 99, 100). It needs to be corrected throughout the manuscript.

Response: We thank the reviewer this comment and have updated the grammar and references.

Participants section: I propose to add years of training experience and a category in the disability classification for Rugby.

Response: We thank the reviewer this comment. As the data has been collected by Wheelchair Rugby Canada and we have been allowed to use anonymized data we have not been given this information and cannot connect this information back to the data but we have included what we can.

Line 119 – What is the symbol next to Xsens?

Response: We thank the reviewer this comment and have updated the symbol.

Lines 139 – 140 – there should also be an explanation of θ.

Response: We thank the reviewer this comment and added an explanation.

Section 2.4.3 Resistance Force Modelling should be divided into two chapters, AF and CRF, because otherwise the formulas blend together.

Response: We have added two chapters to help distinguish the chapters.

Major comments:

Results section should be refined. I don't understand subsection 3.1 What does Figure 2 refer to? There is no reference in the text. Is this for one person? Please write using what equation the results were obtained for velocity modeling. I understand that it was equation 2a?

Using what test were the results described in Table 1 compared? What is the FV slope, and DRF? How were they counted? How was theoretical maximum force (?0), and maximum power output counted? This are information that should be described in the methods. Figure 4 is important and should be properly discussed in the text. Please also number the subplots as A, B, C and D. In the description of the figure there is a reference to the work of Janssen et al. This was not mentioned in the methods.

In conclusion, the authors need to describe the chapters material and methods and results more precisely so that they are compatible. The reader must have clarity on what results from what and what tests are being compared.

Response: We truly thank the reviewer for these valuable major comments, we have updated and refined the results and methods based on these extensive and valuable comments. We believe that these modifications truly improve the clarity of the document.

Round 2

Reviewer 1 Report

The revised version of the manuscript has been improved.